# A Methodology for Measuring Actual Mesh Stiffness in Gear Pairs

Carlo Rosso [1,2,*], Fabio Bruzzone [1,2], Domenico Lisitano [1] and Elvio Bonisoli [1,†]

[1] Department of Mechanical and Aerospace Engineering, Politecnico di Torino, 10129 Torino, Italy; fabio.bruzzone@polito.it (F.B.); domenico.lisitano@studenti.polito.it or domenico.lisitano@polito.it (D.L.)
[2] GeDy TrAss s.r.l., 10128 Torino, Italy
[*] Correspondence: carlo.rosso@polito.it
[†] Dead prematurely on 2023.

**Abstract:** The measurement of the meshing stiffness in gear pairs is a technological problem. Many studies have been conducted, but a few results are available. A tailored test bench was designed and realized to measure the Static Transmission Error in two mating gears to address this issue. The bench is capable of testing several kinds of gears, e.g., spur, helical, conical, and internal, and it measures the transmission error concerning the applied torque. The Static Transmission Error is due to the variable stiffness of the gear teeth during a mesh cycle. In this paper, a dynamical method for measuring gear mesh stiffness is presented. The tooth stiffness is estimated from the torsional modal behavior of the rotating parts of the test bench. The dynamics of the system are acquired using accelerometers and very precise encoders to measure the angular accelerations and displacements of rotating parts. The torsional mode shapes are identified; those that show a vibrational behavior of the gears that do not follow the transmission ratio's sign of the mating kinematic condition are selected because they depend on the flexibility of the teeth. In such a way, the engagement stiffness is estimated from the natural frequencies of the selected mode-shapes and the known inertia of gears and shafts. The experimentally identified results are also compared with numerical values computed with a commercial software for mutual validation.

**Keywords:** gears; mesh; dynamic stiffness; experimental; uncertainty

## 1. Introduction

Geared transmissions are widely used in mechanisms to transmit motion and power in a variety of fields. One of the main sources of possible dynamic overloads or Noise, Vibration and Harshness (NVH) problems is considered to be the Static Transmission Error (STE) [1]. By definition, the STE is the difference between the angular position under load in a pair of engaging gears and the theoretical one. This position difference is mainly due to the tooth stiffness, so the STE and the meshing stiffness are the two sides of the same coin. In the present paper, the STE and meshing stiffness can be considered the reciprocals of each other. The STE is the main driver of several important phenomena which are typical of geared transmissions, and experimental values are important for several simulation methodologies. This key parameter has been widely studied and modeled using a plethora of different models, starting from the analytical work of Weber [2] and the experimental works conducted by Harris [3]. However, experimental measurements are scarce. Finite Element (FE) models have been widely used [4–6], but they are generally cumbersome in both the generation of the model and in the solution phase, yet they are very precise. Recently, a semi-analytical model was proposed [7,8] which couples the speed of the analytical solution with a nonlinear solution scheme and a non-Hertzian contact model, and it is capable of matching the accuracy of the FE models with a greatly improved ease of use in 2D and 3D cases. In the history of the field, several test benches have been developed [9–11] and proposed, but their attention is mostly focused on the

measurement of the Transmission Error (TE) in working conditions under torque and rotational speed, which is also commonly called the Dynamic Transmission Error (DTE). Recently, some authors have investigated the possibility of measuring the STE directly on gears. In [12], a test bench was developed for measuring the STE using an optical evaluation of the tooth deflection. The FZG (FZG = Forschungsstelle für Zahnräder und Getriebebau, Technische Universität München (Gear Research Centre, Technical University, Munich), Boltzmannstraße 15, D-85748 Garching, Germany) [13] test bench configuration is typically employed for testing gear pairs, and several measurement devices are implemented. For example, in [14], the application of an accelerometer on a gear tooth is proposed for estimating the natural frequency of gear meshing. Sometimes, the meshing stiffness variation is considered to evaluate any possible damage in the gears, and some authors use the measure of gear meshing stiffness for monitoring purposes, but direct or indirect measurements are not available for different types of gears. For example, in [15–18], the same test rig with a simplified gear pair was used with different measurement techniques—strain gauges, photoelasticity, laser displacement sensor and experimental modal analysis (EMA)—in order to estimate the meshing stiffness with particular emphasis on the presence of a damage. In [19], strain gauges and magnetic pickups were used to measure the stiffness of the tooth in a test bench with a standard configuration. In [20], an FZG configuration and encoder signals were used to evaluate the Dynamic Transmission Error and STE; a similar logic, but implemented on a more flexible test bench, is described in [21]. Modal analysis was also used on a simple test rig made of two shafts and a gear pair, and the results were compared to the estimation of the finite element model [22]. Dynamic stresses, which can be correlated with gear stiffness, can be measured in several ways: strain gauges [23] and fiber Bragg gratings [24]. Usually, the measurements are performed directly on a gearbox and not on a pair of gears, such as in [25,26], but sometimes, the tests are performed on a single tooth, such as in [27]. By combining the experiences listed above and trying to overcome some limitations of the previous configurations, such as the possibility to test any possible gear shapes (cylindrical, internal and bevel), a new test bench was developed and built [28] in Politecnico di Torino. In this paper, this test bench is described in detail, and a combination of EMA and encoder signals are used to propose a measurement technique for the Static Transmission Error and, hence, the tooth stiffness. The test bench is composed of two turrets, one of which is movable to allow for the accommodation of different gear sizes and types. Internal and external gearings can be tested as well as, such as bevel or hypoid gears. The measurement is generally driven by the motion of two counter-moving racks of weights, which generate the driving load at the same time. In this study, the said weights do not move but are kept fixed, and they only generate the mean torque. This particular configuration allows us to overcome some limitations:

- Different kinds of gear pairs can be tested with the same equipment and until the limit of 500 Nm of torque;
- Runout and No Load Transmission Error do not affect the measurements of the tooth stiffness;
- Single or double contact (for gear pairs with a contact ratio lower than 2), $n - 1$ or $n$ contact (for contact ratio higher than $n$) configurations can be tested separately.

In this paper, the engagement is studied only under two conditions: one, when only one pair of teeth is in contact; two, when two pairs of teeth are engaged at the same time. The measurement details the stiffness of the teeth under dynamic conditions by modifying the test bench with an exciting shaker and obtaining the data with accelerometers coupled with high-precision rotary encoders. The results are presented at different torque levels at the two meshing positions, and they are compared with the semi-analytical predictions included in a dynamic model of the studied test bench.

## 2. Test Bench Description

The test bench was designed to measure the STE in a couple of engaging gears. The target of the test bench is to experimentally evaluate the variation of the angular

position during meshing [1]. In order to obtain such a measure, two gears are driven on the test bench by very slow-moving loads. The two gears are mounted on two different supports to allow for the engagement of different kinds of gears, ranging from spur to bevel and from external to internal.

Figure 1 shows the whole setup that can be summarized into five main groups of components:

- Structural parts;
- Transmission of the motion;
- Measurement system;
- Security system;
- Tested gears.

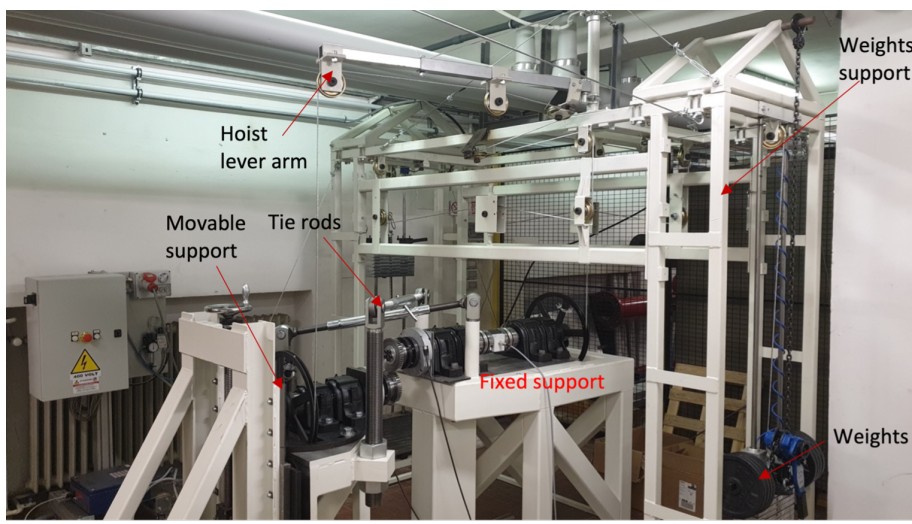

**Figure 1.** Picture of the used test rig.

The structural parts have to enable the transmission of the motion and host the measurement system, with the strict target of maintaining high stiffness and flexibility in all possible engagement configurations. The test bench provides input torque and a braking torque through the moving loads. The measurement system records the angular position of both the gears and the torque involved in the engagement via rotary encoders and a torque meter.

*2.1. Structural Parts*

The structural parts are as follows:

- Weights support;
- Fixed support;
- Movable support;
- Fixed platform.

The weights support consists of a frame in which two sets of "gym" weights can move up and down. Because of the sense of rotation, the two sets of weights alternatively provide driving and resistant torques. The loads generate forces that, through a series of pulleys, are delivered by two steel cables to the two main pulleys connected to the gears. In particular, the upper part of the support has a hoist lever arm that can transfer the braking/driving torque from the weight support to the braking/driving shaft on the movable support. This arm can be easily adapted to the position of the movable support through a rotation and through two movable brackets. The fixed support is composed of rigid metal profiles on which a shaft group is located. In addition, the fixed support is doweled to the through a 10 mm thick plate. The movable support, on which the other shaft group is located, can be shifted for the setup over a metallic platform clamped to the ground. Once the movable support is moved in the correct position for the engagement, the tie rods are mounted to connect the fixed and movable supports directly in order to close the force loop. Finally,

the movable support is then fixed to the metallic platform through two electro-permanent chucks that apply 100,000 N each in the vertical direction and can resist up to 20,000 N in the tangential direction.

Movable Support

The movable support is the most complicated component of the whole test bed. Once the test gears are installed on the two ends of the two shafts, the movable support is moved through a pallet truck to the position that can guarantee the correct engagement. Since the movable support can be carried around through the fixed one, it enables engagement for different types of couplings. For instance, if the two shafts are both parallel, cylindrical gear couples can be tested. If the movable shaft axis has an angle with the corresponding axis of the fixed shaft, bevel gears can be tested. Once the macro-adjustment is performed, micro-adjustments are carried out using the different mechanisms installed on the movable support. The permitted micro-adjustments are linear displacements along approaching and vertical directions, as well as rotations around vertical and horizontal axes. The output shaft is mounted on a rotating platform, as shown in Figure 2.

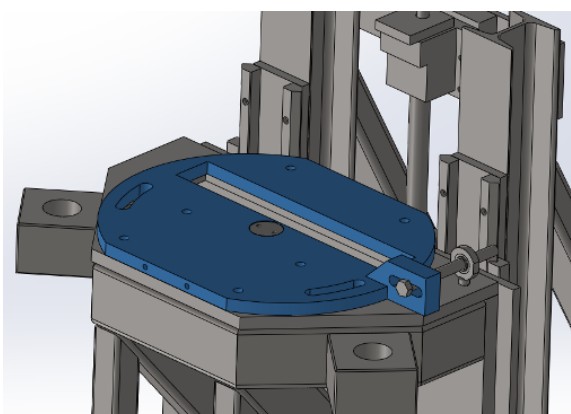

**Figure 2.** Plate for rotational micro-adjustments.

This platform can rotate owing to an adjusting screw around the vertical axis. This regulation can be used to eliminate misalignment or to see the effect of an imposed misalignment on the STE. The rotating platform can be moved in the vertical direction via a trapezoidal threaded spindle mechanism controlled through a handwheel as depicted in Figure 3.

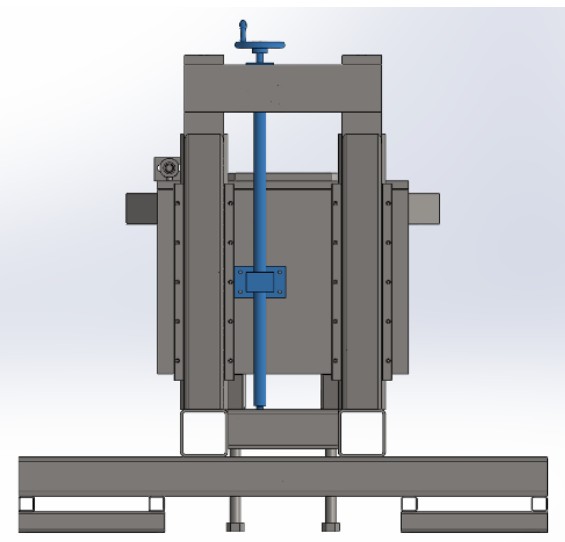

**Figure 3.** Trapezoidal threaded spindle mechanism for the vertical adjustment.

This regulation can be used to see the effect of a higher center distance of the shafts on the contact ratio and, therefore, on the STE. Displacements along the approaching direction and rotation around the vertical and horizontal axes are guaranteed by four threaded pins positioned at the vertices of a rectangular vertical plate.

### 2.2. Transmission of the Motion

This transmission is guaranteed by the following sub-groups of components:

- Pulleys and brackets;
- Shaft pulleys.

Once gears are properly mating each other according to the theoretical engagement, the weights dedicated to the input torque have to be lifted through an electric hoist, whereas the braking weights have to lie at the bottom. When the test is running, the input torque weights descend in a controlled manner, while the braking weights start ascending. Each group of masses possesses a total amount of 125 kg, discretely selectable from 5 kg to 125 kg in 5 kg increments. To overcome friction and losses, some additional loads are hanged to the tailored appendices. The weights are lifted using a 5 mm diameter steel rope that passes through a series of 110 mm diameter pulleys. All the pulleys installed on the weight supports are made of steel, integrating a ball bearing. The path of the forces for the braking torque runs through the ropes and is represented in Figure 4, in which the maximum load of 125 kg is lifted. Due to the gravity, the maximum force driven by loads is about 1250 N. This force can then be doubled by means of a guide pulley that moves horizontally on the weight support: this means that with the maximum weights it is possible to apply a tangential force of 2500 N at the extremities of the shafts where two identical pulleys are fixed. Their diameter is 400 mm in order to provide a maximum input torque of 500 Nm.

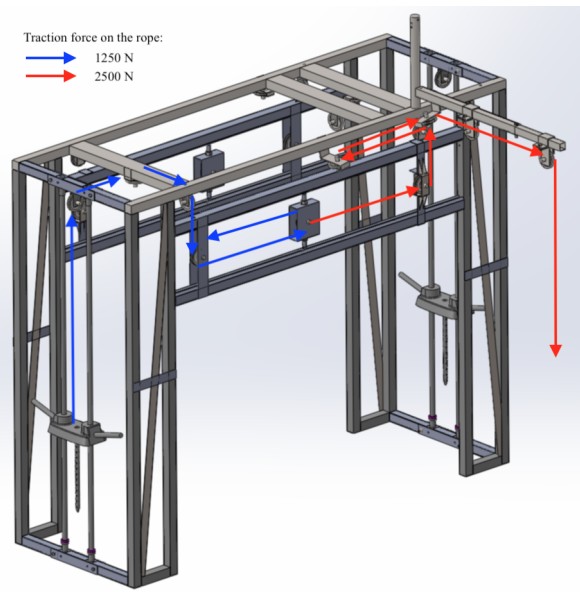

**Figure 4.** Weight support with rope pathways and force directions.

### 2.3. Measurement System

The measurement system is composed of three main devices, two encoders for evaluating the angular displacements of the two engaging gears and a torque meter for assessing the actual amount of torque transmitted by the gears. The torque meter is a T22/500 Nm from HBM GmbH that only detects the torque value without any retroactive feedback on the weights. Two Heidenhain RCN8580 encoders, shown in Figure 5, are mounted through a mechanical joint that transmits to the encoder only the rotational component of the gear shaft displacements. They are absolute angle encoders with a 1″ accuracy and measure standard DIADUR circular scales with an absolute and incremental track of 32,768 lines

per revolution. Owing to the EnDat 2.2 interface, it is possible to obtain 536,870,912 (29 bits) position values per revolution.

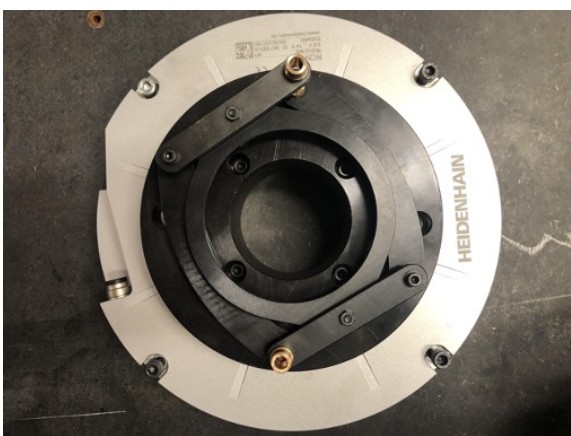

**Figure 5.** RCN encoder and the decoupling mechanical joint mounted on it.

The transmission error is acquired during the motion of the weights at different torque values. It is recorded as the difference between the two encoder signals. In Figure 6, a typical trend of an acquired Transmission Error (TE) is shown. In order to extract the portion of TE due to the compliance of gear contact, a tailor-made technique was developed. In Figure 7, the evaluation of the Peak-to-Peak TE (PPTE) due to gear compliance is compared to the one computed using the GeDy TrAss software [7,8,29,30].

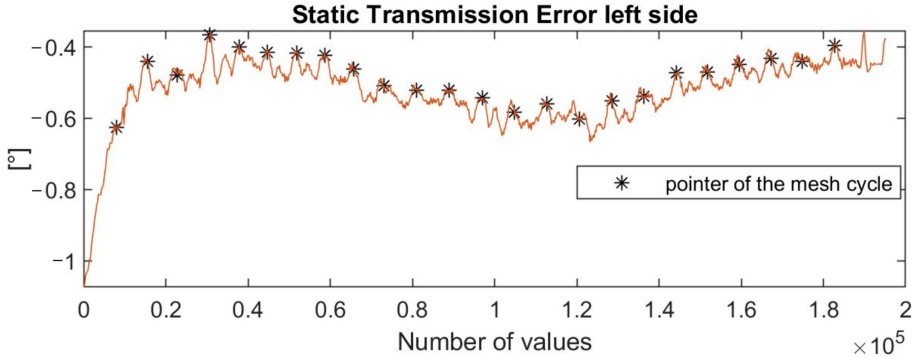

**Figure 6.** Example of an acquired STE; the stars indicate the tooth engaged.

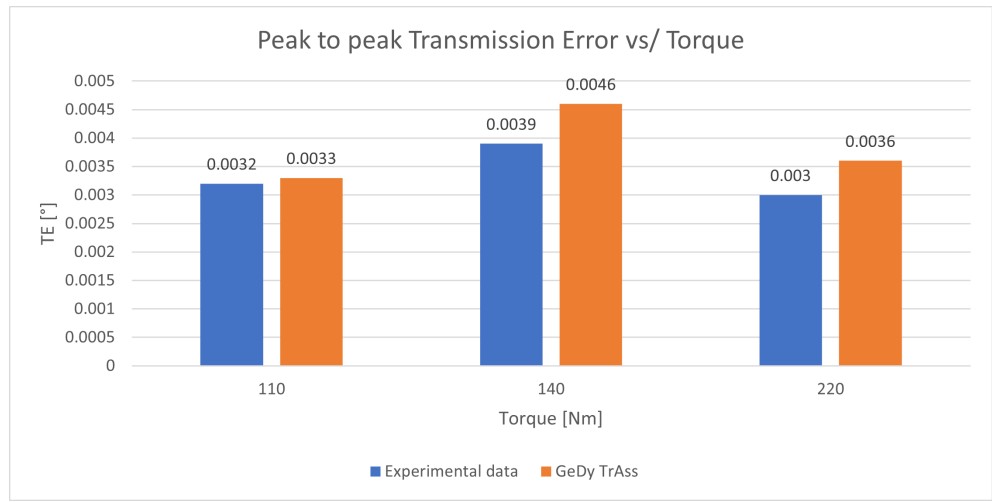

**Figure 7.** Comparison between measured and calculated PPTE.

*2.4. Tested Gears*

The gears used in this paper are a copy of the one proposed in [31], and the technical data are summarized in Table 1:

**Table 1.** Data of the tested spur gears.

|  | **Gear 1** | **Gear 2** |
|---|---|---|
| Number of teeth [-] | 28 | 28 |
| Modulus [mm] | 3.175 | |
| Face width [mm] | 6.35 | |
| Gear body width [mm] | 6.35 | |
| Pressure angle [°] | 20 | |
| Helix angle [°] | 0 | |
| Contact ratio | 1.638 | |
| Rack | ISO 53/A [32] | |
| Maximum input torque [N mm] | 101686 | |
| Material | AISI 1040 | |
| Young's modulus [GPa] | 207 | |
| Density [$\frac{kg}{m^3}$] | 7800 | |

## 3. Motivation of the Work

A better understanding of the contribution of each element to the TE was the goal of the study reported in this paper. During rotation, the No Load Transmission Error (NLTE) and Runout Errors have to be removed from the acquisition to properly estimate the STE. Some data analysis procedures can be applied, such as the one used in [20]. In this present study, the idea under investigation was to remove the relative motion between gears. By considering the static condition of equilibrium during each position in the mesh cycle, the dynamic properties are only related to the actual gear stiffness. The idea developed in this study is to freeze the system in a precise angular position (i.e., single or double teeth contact for the particular type of gears) and, acting with a variable ripple of torque, to acquire the dynamic response of the gears. By comparing the data coming from the accelerometers and the encoders, it is possible to identify the resonance condition due to the gear mesh. The measured frequency can be correlated with the gear stiffness through the inertial data. A basic model of the test bench, in which the inertia and the stiffness of the main elements are introduced, is used and the stiffness of the engagement and can be evaluated by tuning it until the numerical and experimental data match. The meshing stiffness is directly related to the Static Transmission Error due to gear compliance, according to the equation:

$$k_m = \frac{T}{STE \cdot r_b} \tag{1}$$

where $r_b$ is the base radius of one gear, and $T$ is the torque on the gear. A commercial software (GeDy TrAss) is used to compute the STE and, hence, the meshing stiffness to speed up the tuning process.

## 4. Proposed Methodology

To confirm the idea expressed above, the test bench was equipped with a shaker (Dongling ESD-045 with a maximum force of 450 N in the 5 ÷ 3000 Hz range) hanged to the weight support through an elastic frame. The stinger of the shaker was then connected to a load cell to measure the tangential force applied to one of the driving pulleys. Then, the load cell was connected through a tailored support to the pulley to transform the tangential force provided by the shaker in a torque ripple on the gears. In Figure 8, the excitation system is shown. An additional accelerometer was coupled to the pulley to acquire the tangential acceleration imposed by the shaker. A spherical joint was inserted between the load cell and the pulley adaptor in order to guarantee the tangential action of the force; once the

gears were positioned in the required position, the device was adjusted, and the spherical joint was tightened using a screw to avoid play and, hence, additional nonlinear effects. Two triaxial accelerometers (PCB 356A15 with a sensitivity of 100 mV/g in the 2 ÷ 5000 Hz frequency range) were then connected to the gear and, in particular, acceleration was recorded in the tangential direction to estimate the rotational acceleration of the gears. In Figure 9, the two accelerometers can be seen, and in Figure 10, the final layout of the test bench can be seen. To ensure the target level of torque, the weights were correctly engaged on the sliders, and the motion was locked through the pneumatic blocks installed on each slider. In such a way, the torque meter registered a constant torque with the shaker turned off, and the reciprocal angular position of the gears was frozen. By activating the shaker, the torque meter registers an oscillating torque across the constant-value set through the weights.

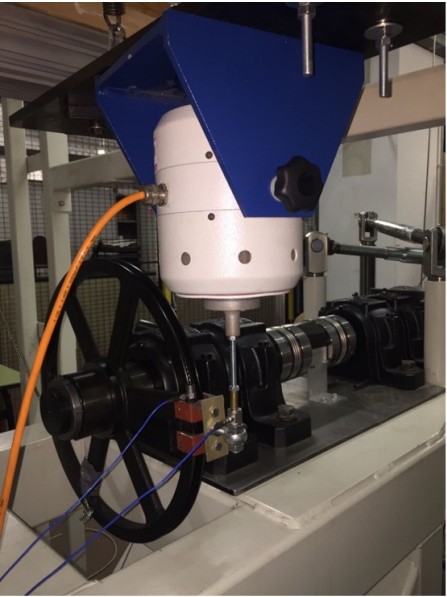

**Figure 8.** Connection device to transform the tangential force generated by the shaker in torque ripples on the driving shaft.

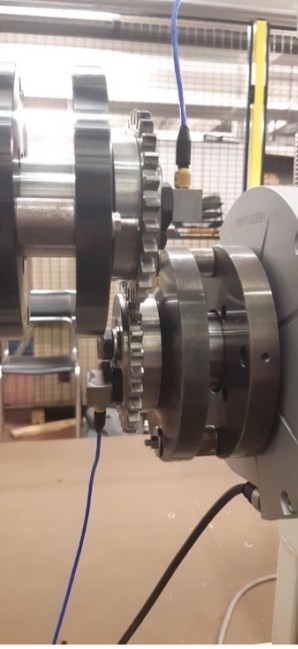

**Figure 9.** Triaxial accelerometers applied on the gears.

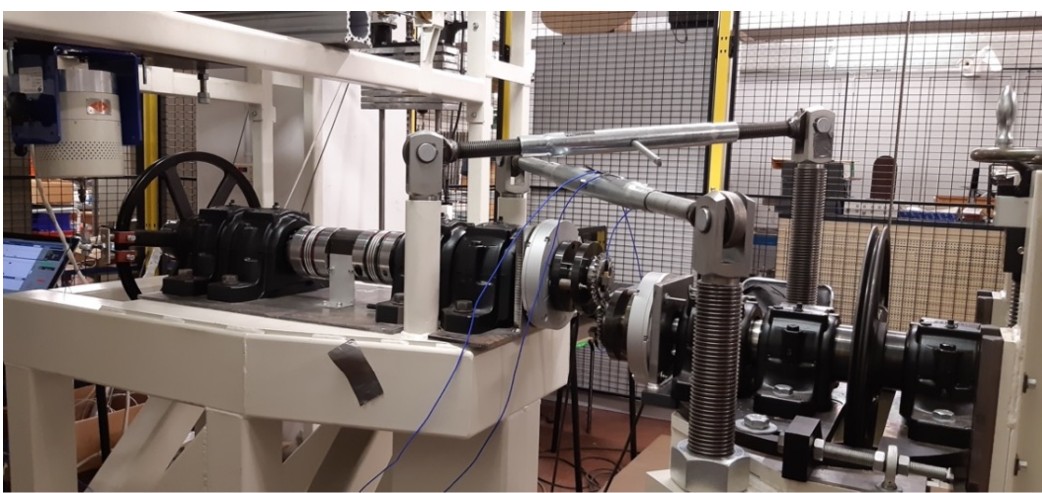

**Figure 10.** Test bench equipped as described.

## 5. Experimental Results

Two angular positions were analyzed: one that depicts a single pair of teeth in contact, and one that represents the condition of two contact pairs. These effective conditions were confirmed through a feeler gauge (0.05 mm). These two positions were far apart enough such that the contact pattern remained the same with the torque application. In Figure 11, the two positions are shown. Different torque ripples at different mean torque levels were tested, and both the accelerations and angular positions of the gears were acquired. In particular, the tests were conducted at 50, 70 and 90 Nm of constant torque and three values of torque ripple: 4, 5 and 9 Nm. Frequency sweeps were performed from 20 to 2000 Hz at 5 Hz/s increments to first analyze the behavior of the test bench. Then, once the frequency range of interest was selected (200–450 Hz), the sweeps were performed across this range with an increment of 0.5 Hz/s. In Figures 12 and 13, examples of the acquired signals are reported. To establish the proper frequency range of interest, the encoder signals of both gears were analyzed.

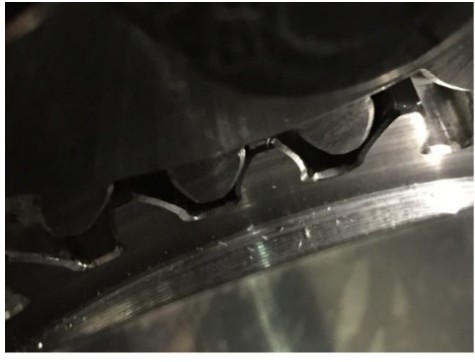 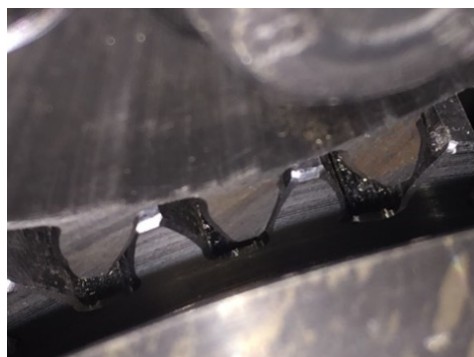

(**a**) Single-contact position. (**b**) Double-contact position.

**Figure 11.** Positions used to estimate the gear mesh stiffness.

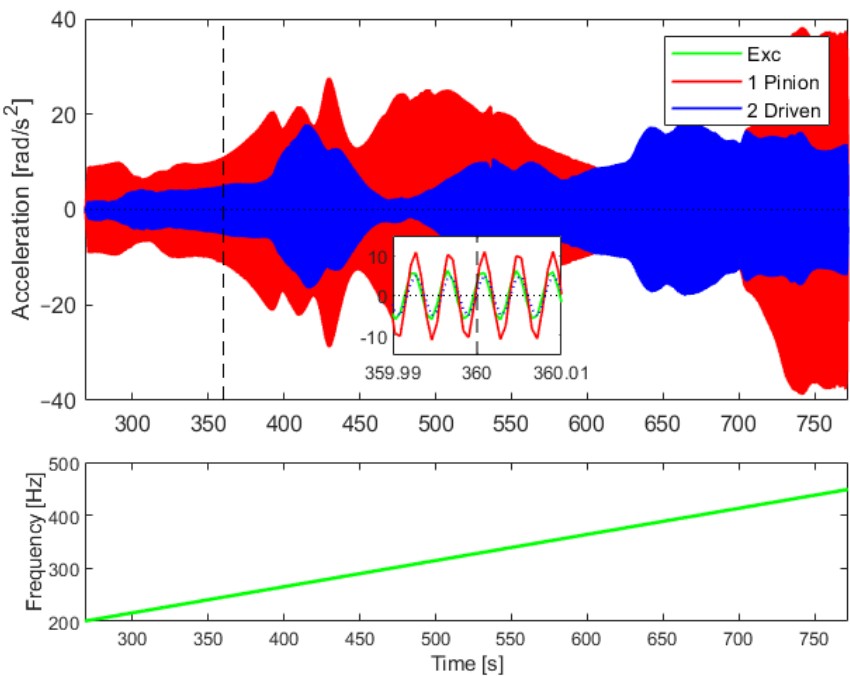

**Figure 12.** Frequency sweep and acceleration output recorded by accelerometers.

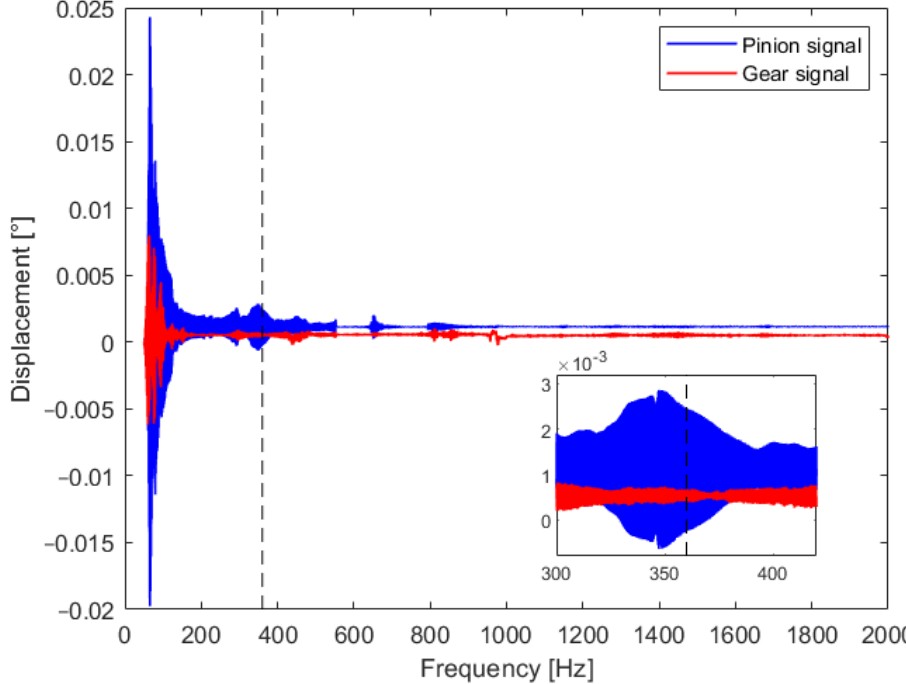

**Figure 13.** Encoder angular displacement output recorded during frequency sweep (the same as Figure 12).

During normal working conditions, two external gears have opposite rotations due to their kinematics. The two encoders used in this test bench were mounted face-to-face, so the acquired signals in normal working conditions were synchronous because the positive sense of rotation was the same for the two gears (see Figure 14).

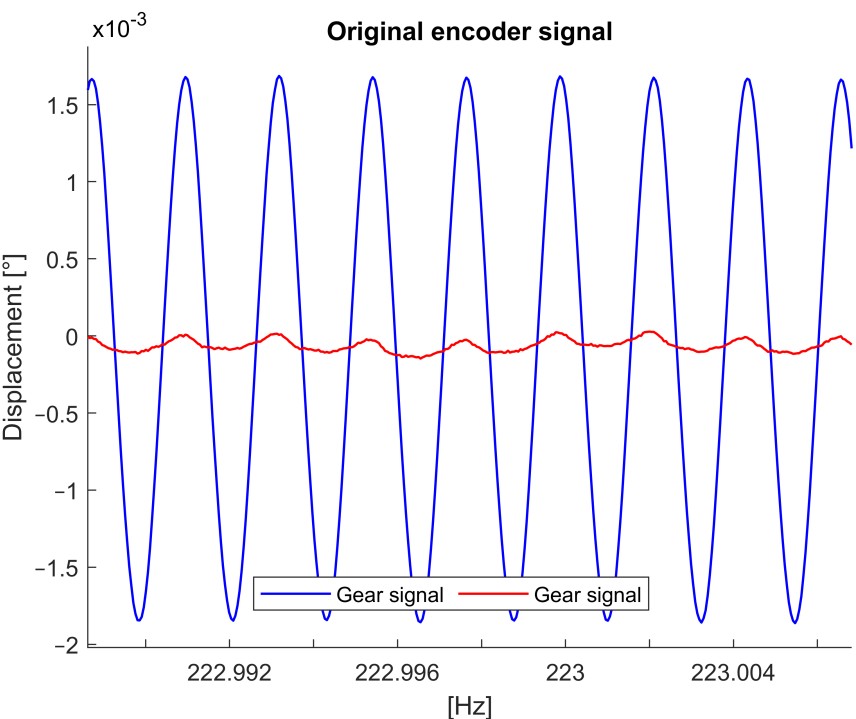

**Figure 14.** Synchronous angular displacement signals between two gears.

In meshing resonance, corresponding to a non-kinematic relationship of the gear motion, the angular displacements of the two gears, recorded by the encoders, become opposite. This means that in the acquisition of the encoders, the out-of-phase condition refers to a potential non-kinematic relationship between the two gears. In the range 20–2000 Hz, only one range at 360 Hz for a double-pair contact and at 270 Hz for a single-pair contact presented this out-of-phase condition, as shown in Figure 15a,b. These two frequencies were identified to be the resonance frequencies of the mating gears.

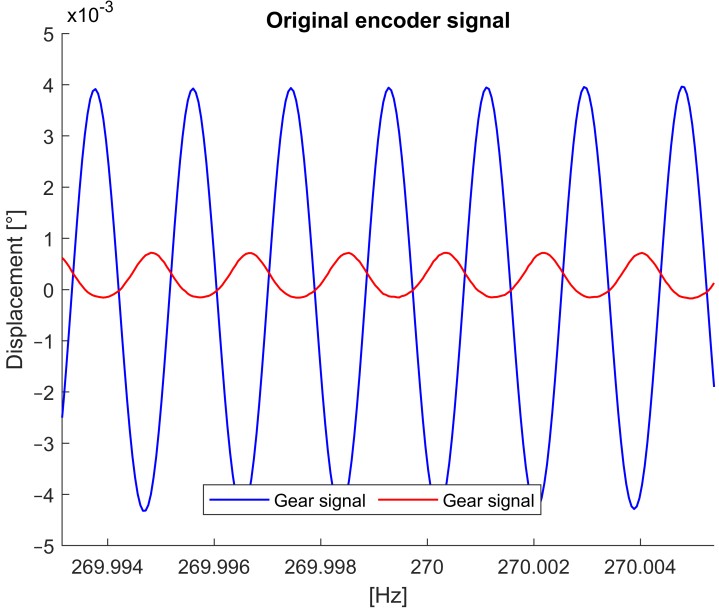

(**a**) Single-contact position resonance at 270 Hz with 90 Nm torque

**Figure 15.** *Cont.*

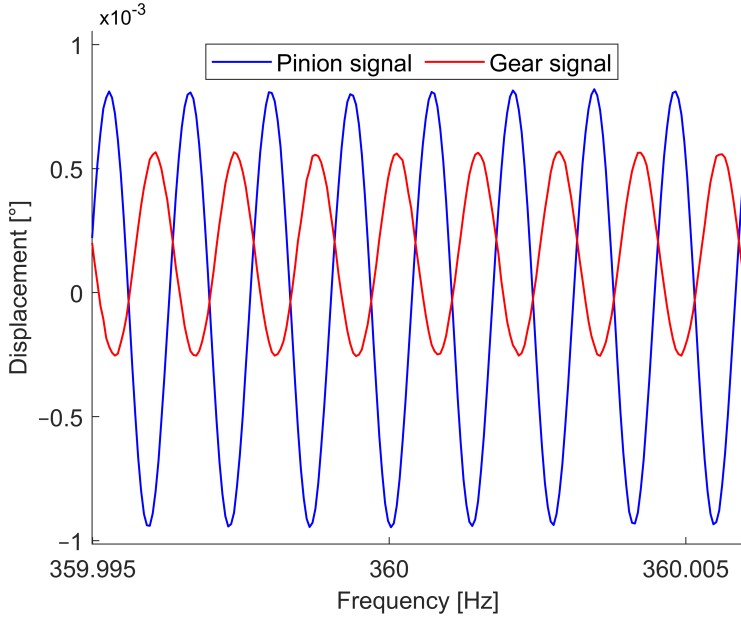

(**b**) Double-contact position resonance at 360 Hz with 90 Nm torque

**Figure 15.** Opposition of angular displacements between the two engaged gear signals in (**a**) single-pair contact and (**b**) double-pair contact at 90 Nm of constant torque and 10% of oscillating torque.

## 6. Methodology Discussion

A simplified 1D model was built to further understand the test bench behavior by using the LUPOS approach [33]. The model is quite simple: because no contact losses are expected in the gear meshing, a linear lumped torsional model can be efficient and accurate enough to analyze the test bench. The rotational inertia of each element (two driving pulleys, dofs 1 and 6; two gears, dofs 2 and 3 and the two joints across the torque meter, dofs 4 and 5) was obtained from the CAD model of the test bench, and the torsional motions of these elements were the computed and measured degrees of freedom of the model. A simple linear (gear meshing, cables) or torsional stiffness (shafts and joint) connects the 6 degrees of freedom. The stiffness of the cables that connect the pulleys to the ground was estimated using measures of displacements under the load of the weights. The stiffness of the joints connecting the torque meter was set to the value provided by the manufacturer. The meshing stiffness was assumed according to the calculation performed with the GeDy TrAss software and used to connect the input and output shafts as in a block-oriented approach [34]. The meshing stiffness was estimated at different torque levels with the approach shown in [7]. In Figure 16, a sketch of the model and the CAD are shown. The resulting model was a 6 torsional-degrees-of-freedom lumped parameter system according to previous assumptions and according to the experimental measurements made on the test bench; in Figure 17, the modal identification performed in Siemens TestLab is shown. An experimental test on the test bench performed in the range of 20–2000 Hz identified the natural frequencies. The frequency response of the test bench regarding torsional modes confirmed the data evaluated using the simplified model.

The numerical modal analysis was performed using the presented 6 dof model: normal modes and natural frequencies were estimated for several mean torques and both meshing conditions, and they were compared with the experimental acquisitions. In Figure 18, the inertance of the driven gear with respect to the applied torque ripple in a single contact at three levels of the mean torque is presented. It is possible to highlight the frequency shift due to the increase in the mean torque. The increase in torque is related to the increase in the meshing stiffness, and both the numerical model and the experimental test provide the same trend. A comparison of the results is presented in Table 2.

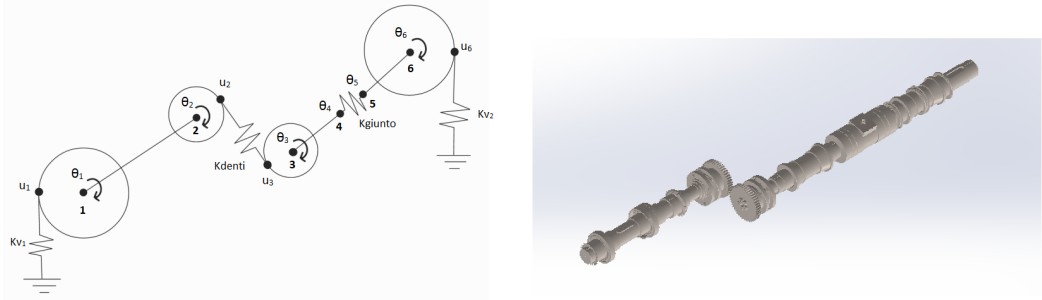

**Figure 16.** Comparison of the numerical model and the CAD model.

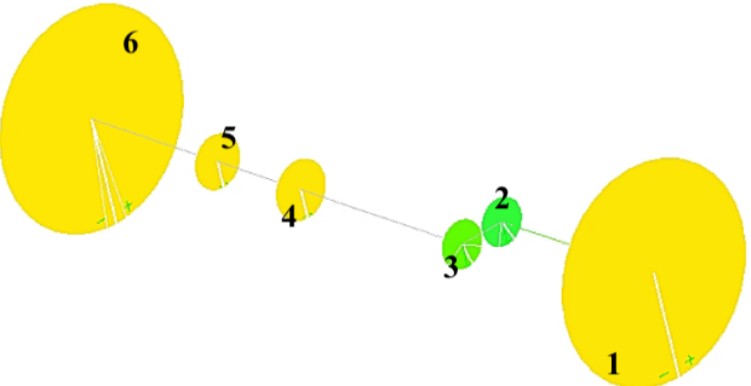

**Figure 17.** Identification process of the experimental setup performed in Siemens TestLab; numbers correspond to the evaluated dofs, in yellow the additional inertia, in green the two tested gears.

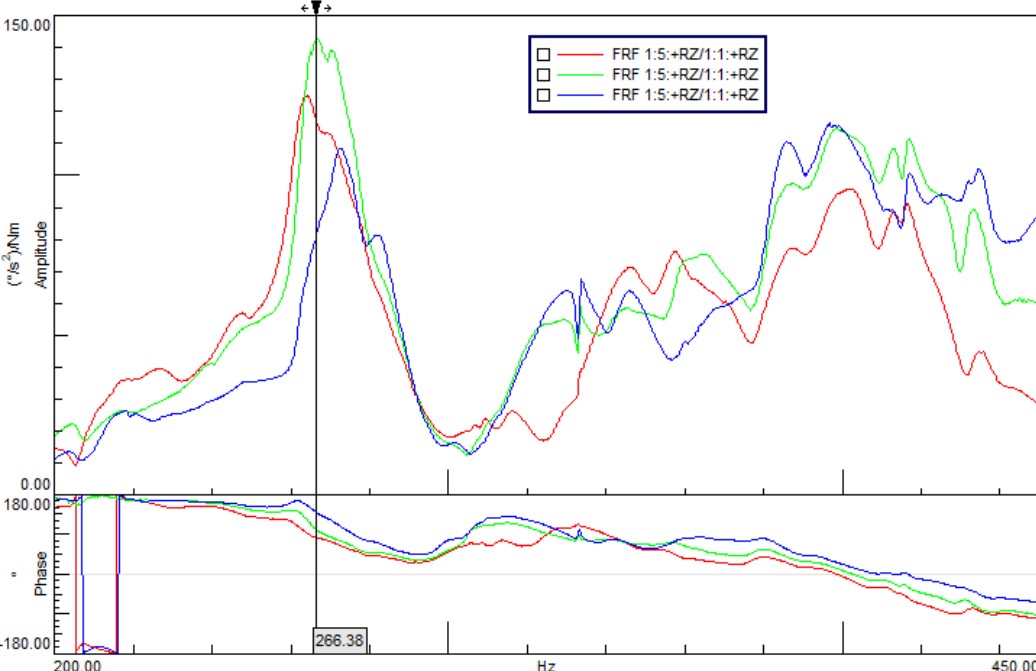

**Figure 18.** Example of a frequency shift for the single-contact pair around the meshing resonance.

In Tables 2 and 3, the results of the numerical model are summarized. By observing the frequencies and the mode shapes, the second natural frequency is the one with the higher dependency on mesh stiffness variation; in addition, the second normal mode presents a non-kinematic behavior of the two modal coordinates corresponding to the gear rotational motions (points 2 and 3 in the model of Figure 16 and in the modal identification of Figure 17). If the motion is kinematic, the displacements must be opposite, so the behavior

of the gears reflects the correct engagement; whereas the torsional displacements are in the same direction, and the gear teeth are moving out of contact, depicting a non-kinematic movement. This condition is also highlighted by the fact that the second mode shape is the only one in which all the degrees of freedom have the same directions. Obviously, this is contrary to the standard displacement of the system. As a matter of fact, in a correct motion, nodes 1 and 2 have a rotational direction opposite to that of nodes 3, 4, 5 and 6 due to the engagement of two external gears.

**Table 2.** Natural frequencies with respect to mesh stiffness and torque variations. In grey the natural frequency referred to the meshing resonance.

|  | Single Pair Contact | | | Double Pair Contact | | |
|---|---|---|---|---|---|---|
| Torque [Nm] | 50 | 70 | 90 | 50 | 70 | 90 |
| Mesh stiffness [kN/m] | 2.085 | 2.31 | 2.33 | 3.57 | 3.97 | 4.01 |
| Mode 1 Freq. [Hz] | 53 | 53 | 53 | 53 | 53 | 53 |
| Mode 2 Freq. [Hz] | 293 | 305 | 306 | 355 | 367 | 368 |
| Mode 3 Freq. [Hz] | 799 | 802 | 801 | 810 | 813 | 813 |
| Mode 4 Freq. [Hz] | 952 | 958 | 959 | 1001 | 1015 | 1016 |
| Mode 5 Freq. [Hz] | 1285 | 1286 | 1286 | 1294 | 1298 | 1281 |
| Mode 6 Freq. [Hz] | 20,413 | 20,413 | 20,413 | 20,413 | 20,413 | 20,413 |

**Table 3.** Mode shapes for single and double contacts @ 90 Nm. In grey the modal coordinates of the two gears and in yellow the mode referred to the meshing frequency.

| $\eta$ | Mode Shapes for Single-Pair Contact | | | | | | Mode Shapes for Double-Pair Contact | | | | | |
|---|---|---|---|---|---|---|---|---|---|---|---|---|
|  | 1st | 2nd | 3rd | 4th | 5th | 6th | 1st | 2nd | 3rd | 4th | 5th | 6th |
| 1 | 3.29 | 4.60 | 4.54 | −1.00 | 0.08 | ≈0 | 3.29 | 4.91 | 4.14 | −1.21 | 0.16 | ≈0 |
| 2 | 3.336 | 3.25 | −5.24 | 2.09 | −0.37 | ≈0 | 3.34 | 2.76 | −5.03 | 3.00 | −0.77 | ≈0 |
| 3 | −3.36 | 2.08 | 1.62 | 5.58 | −3.28 | −0.01 | −3.36 | 1.68 | 2.34 | 5.16 | −3.73 | −0.01 |
| 4 | −3.36 | 2.99 | −0.77 | −0.48 | 4.13 | 4.69 | −3.35 | 3.05 | −1.04 | 0.23 | 4.06 | 4.69 |
| 5 | −3.36 | 2.99 | −0.78 | −0.50 | 4.12 | −8.11 | −3.35 | 3.05 | −1.04 | 0.21 | 4.04 | −8.11 |
| 6 | −3.32 | 3.26 | −2.16 | −6.25 | −6.20 | 0.02 | −3.32 | 3.49 | −3.05 | −6.12 | −5.83 | 0.02 |

## 7. Conclusions

This experimental study highlights that in the range of interest, the vibration of the system around the equilibrium condition presents just one occurrence in which the two gears are not in kinematic accordance. This condition is highlighted by the non-synchronous motion registered by the two encoders mounted on the gears. In other words, the dynamic response of the test bench cannot only be described by the accelerometers, but the information coming from the encoders is mandatory; otherwise, it is not possible to properly identify the meshing resonance. The proposed methodology, hence, proves to be effective in recognizing the resonance condition due to meshing stiffness because by combining the accelerometer and encoder measurements, it is easy to identify the non-kinematic condition responsible for the meshing resonance, which, otherwise, is not possible. A second important aspect is that the methodology allows us to estimate the meshing stiffness as an indirect output. The meshing stiffness can be evaluated on the basis of the meshing resonance frequency and the inertia characteristics of the test bench. The methodology proves to be effective in estimating the stiffness value of the engagement under different loading and mating conditions independently from the Runout or Non-loaded Transmission Errors. In addition, it is easy to highlight that the resonance peak increases in frequency with the increase in torque. By comparing the numerical and experimental data, it is also possible to state that the numerical model and the experimental setup agreed in the identification of a resonance condition due to the gear mesh stiffness. The value of this mesh stiffness was computed using the GeDy TrAss software with respect to the engagement conditions. In particular, two values of mesh stiffness were computed: one for the single-contact pair and another for the double. These data were used in a

6 dofs lumped parameter model to compute the dynamic behavior of the test bench, and the experimental outcomes and numerical results matched with a high accuracy. With further analyses, the proposed methodology can be used to conduct an accurate study on the effects of structural damping; for example, a measurement of the damping occurring in the meshing can be carried out as in [35]. Another possible use of this methodology is related to the effect of the lubricant, in particular to experimentally measure the stiffness and damping components introduced by the lubricant film.

**Author Contributions:** Conceptualization, C.R. and F.B.; methodology, C.R., F.B., E.B. and D.L.; experimental test, C.R., F.B., E.B. and D.L.; writing—review and editing, C.R., F.B. and D.L. All authors have read and agreed to the published version of the manuscript.

**Funding:** This study was carried out within the Italian Ministerial Decree no. 1062/2021 and received funding from the FSE REACT-EU—PON Ricerca e Innovazione 2014–2020.

**Data Availability Statement:** Data are private, but are available upon reasonable request.

**Acknowledgments:** The authors would like to acknowledge GeDy TrAss s.r.l. for the use of the software and for the possibility to share the obtained results.

**Conflicts of Interest:** The authors declare no conflicts of interest.

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
