# Peer review of "A Methodology for Measuring Actual Mesh Stiffness in Gear Pairs"

_vibration, doi:10.3390/vibration7010011_

Round 1

Reviewer 1 Report

Comments and Suggestions for Authors

â‘ The importance of the problem to be solved should be highlighted in the abstract, where the conclusion is not obvious.

â‘¡Introducing the research methodology in the introduction should indicate the innovative or advantageous nature of the methodology.

â‘¢Figures 15 a and Figure 18 is not clear

â‘£How exactly is a 6-degree-of-freedom lumped parameter system obtained?

⑤Write down the exact value of the natural frequency experimentally tested in Section V.

â‘¥In the conclusion, please include details or data on how the proposed methodology proved to be effective.

Reviewer 2 Report

Comments and Suggestions for Authors

This paper covers a methodology for measuring actual mesh stiffness in gear pair. Reviewer considers that the following revisions are necessary in order to improve the paper.

1. Whole: The originality should be described clearly comparing with cited papers

2. Table 1: The kind of gears and its material should be shown.

3. Figures 13-15: “Displacement” should be shown clearly with figure.

4.Figure 16: How comparison is numerical and CAD model ? It should be described.

5. The consideration should be described in detail.

Comments on the Quality of English Language

Please check.

Reviewer 3 Report

Comments and Suggestions for Authors

The paper is excellent in describing a test bench in order to measure the static transmission error of distinct paired gears. While the introduction and the description of the test bench is of high professionality and excellent in details, the reader gets confused by 

a) showing results in the methodology section

b) not clearly pointing out the STE as the main driver for the paper.

Results section instead falls back in describing a simulation methodology. 

The authors are encouraged to sort the paper clearly in Methods, Results and Conclusion. Further, the  STE as announced as the aim should be clearly presented.  A clear relevance to field cases should be given to keep the paper excellent.

Reviewer 4 Report

Comments and Suggestions for Authors

Dear Authors The Authors should be congratulated for their large-scale undertaking of experimental research on the static error of gear meshing. All the more so because it is a universal method, as it is also suitable for testing bevel gears. The bench test allows you to test gears of various sizes with internal or external gearing. According to the Authors, a slight modification of the station also allows for the experimental determination of gear stiffness. In addition to the complex design of the test rig, the test also requires accurate equipment, such as a torque meter, very accurate two angle encoders, triaxial accelerometers and GeDy TrAss software. Fig. 6 shows the trend of transmission error, and Fig. 7 shows a comparison of PPTE obtained from measurements and calculations. In Figure 10 you can see the complexity of the test bench in question. It is a pity that the authors did not present the mode shape for single and double engagement. It is also a pity that the gear stiffness diagram is not presented, but in the Conclusions the Authors announce further development work. Among other things, research taking into account the influence of lubrication on gear stiffness. I have a question for the Authors why they use very thin gears for dynamics research, as they do for fatigue research.

Round 2

Reviewer 2 Report

Comments and Suggestions for Authors

The manuscript was improved.

Comments on the Quality of English Language

Please check English.

Reviewer 3 Report

Comments and Suggestions for Authors

The paper is excellent. The authors are encouraged for further embedding this work into real field cases.